# RETRACTED: Synthesis and Characterization of Pyridine-Grafted Copolymers of Acrylic Acid–Styrene Derivatives for Antimicrobial and Fluorescence Applications

**DOI:** 10.3390/mi12060672

**Published:** 2021-06-08

**Authors:** Periyan Durairaju, Chinnasamy Umarani, Jothi Ramalingam Rajabather, Amer M. Alanazi, Govindasami Periyasami, Lee D. Wilson

**Affiliations:** 1Department of Chemistry, Thiruvalluvar Government Arts College, Rasipuram 636007, Tamilnadu, India; 2Department of Chemistry, Government Arts College (Autonomous), Salem 636007, Tamilnadu, India; sachuutay@gmail.com; 3Chemistry Department, College of Sciences, King Saud University, Riyadh 11451, Saudi Arabia; jrajabathar@ksu.edu.sa (J.R.R.); pkandhan@ksu.edu.sa (G.P.); 4Pharmaceutical Chemistry Department, College of Pharmacy, King Saud University, Riyadh 11451, Saudi Arabia; amalanazi@ksu.edu.sa; 5Department of Chemistry, University of Saskatchewan, 110 Science Place—Room 165 Thorvaldson Bldg., Saskatoon, SK S7N 5C9, Canada

**Keywords:** pyridine, acrylic acid, styrene polymer, graft polymerization, antibacterial activity

## Abstract

The goal of the present study was to copolymerize 3-(4-acetylphenylcarbamoyl) acrylic acid and styrene using azo-bis-isobutyronitrile (AIBN) as a catalyst. The resulting copolymers exhibited number average molecular weights (M_n_) of 3.73–5.23 × 10^4^ g/mol with a variable polydispersity (PDI = 2.3–3.8). The amide group of the PMA/PSA polymer was used for grafting poly (-styrene-maleic acid substituted aromatic 2-aminopyridine) by the Hantzsch reaction using a substituted aromatic aldehyde, malononitrile, and ammonium acetate. The polymer can emit strong blue fluorescence (λ = 510 nm) and its thermal stability and solubility were enhanced by polymer grafting. Moreover, the polymer showed the fluorescence spectra of the copolymer had a strong, broad emission band between 300 to 550 nm (maximum wavelength 538 nm) under excitation at 293 nm. The Hantzsch reaction yields an interesting class of nitrogen-based heterocycles that combine with a synthetic strategy for synthesis of grafted co-polymer pyridine-styrene derivatives. The as-prepared pyridine-based polymer compounds were screened against Gram-positive and Gram-negative bacteria, where a maximum inhibition zone toward all four types of bacteria was observed, including specific antifungal activity. Herein, a series of pyridine compounds were synthesized that showed enhanced fluorescent properties and antimicrobial properties due to their unique structure and ability to form polymer assemblies.

## 1. Introduction

In recent decades, researchers have shown interest in the synthesis of fluorescent polymers that have potential applications in pharmaceuticals and optical emission devices and utility as sensor probes due to their unique physicochemical properties [1,2]. In particular, the synthesis, characterization, and biological studies of cyanopyridine-based compounds remain topics of sustained interest [3,4]. Among the naturally occurring heterocyclic compounds, the pyridine moiety occupies an important position in biomedicine with antibacterial [4,5], anti-SARS, antiviral, and anticancer activities [6,7,8]. As well, the pyridine moiety enhances the polymer properties and shows excellent biological, hypogeal, and fluorescent properties [6,7]. The Hantzsch reactions provide access to a wide variety of important multifunctional compounds [8], where this prominent multicomponent reaction is known to yield an interesting range of nitrogen-based heterocyclic and organic polymers [8,9,10,11]. The continued interest in the synthesis, characterization, and biological studies of fluorescent polymers [9,10] provides the motivation to design novel cyanopyridine-modified conjugated polymers for various optoelectronic properties in the field of display technology [12,13].

Presently, there are two methods adopted for the preparation of fluorescent polymers, where one strategy involves polymerization of a monomer containing a fluorescent chromophore through a graft-based polymerization [12,13,14]. The second method involves chemical modification of commercially available or synthesized polymers containing multifunctional reactive groups. Poly(styrene-*co*-maleic anhydride) (PMA) is a polymer material prepared by radical polymerization, which has good solubility, biological activity, thermal stability, and good miscibility in various solvents [15,16]. Modification of fluorescent polymers by grafting and graft (co- or homo)-polymerization techniques has received attention from researchers due to commercial applications [12,13,14,15]. These techniques allow for the incorporation of a variety of functional groups in a polymer network [17,18]. Consequently, polymers with cyanopyridine in the main chain would enable an increase of its charge-carrying properties [19]. The synthetic procedure using maleimide polymer-pyridine substitution has not been previously reported. Based on the above approach, it is feasible to synthesize pyridine-containing polymers directly without any protective groups, which reduces the number of steps [20]. The phenomenon in which organic polymer nanoparticles show higher photoluminescence efficiency in the aggregated state than in solution is called Aggregation-Induced Emission Enhancement (AIEE) [21,22,23,24,25]. These graft polymers can act as a candidate material for the development of new antimicrobial agents against different types of bacteria and fungi [22]. In this study, modified polymers of poly (maleicacid-4-acetylarylamide) (MAS), poly (styrene-maleic acid-4-acetylarylamide (PSA), poly(maleic acid-*g*-arylmethoxy-pyridine) (PMAP), poly(maleic acid-*g*-chloroaryl-pyridine) (PMAP1), poly(styrene-maleic acid-*g*-arylchloro pyridine) (PSMP), and poly(styrene-maleic acid-*g*-arylmethoxy pyridine) (PSMP1) were synthesized via the Hantzsch reaction (Figure 1). These polymers were isolated and characterized by thermogravimetric analysis (TGA), differential scanning calorimetry (DSC), and spectroscopic methods. The photo-physical properties of the polymers were evaluated by UV-Vis and fluorescence spectroscopy. All prepared fluorescent polymers were tested for their antimicrobial properties with Gram-positive/-negative bacteria and anti-fungal activity.

## 2. Materials and Methods

### 2.1. Materials 

All required chemicals were purchased from the Sigma Aldrich, Merck, and Alfa Aesar (Indian) chemical companies. For the spectrophotometer, the ^1^H-NMR spectra were recorded on Bruker 400 MHz and 500 MHz (^1^H frequency) instruments. FT-IR and UV–Vis absorption spectra were recorded on a Perkin Elmer LAMBDA 950 spectrophotometer and fluorescence measurements were recorded on Spectra Max Fluorolog-3 at room temperature. Melting points were recorded on Casia-Sigma (VMP-AM) melting point apparatus and were uncorrected. The reaction time was determined using thin-layer chromatography (TLC) on fluorescent silica gel plates F245 (Merck) viewed under ultraviolet (UV) light at 245 and 265 nm. Silica gel (120–400 mesh) was used for column chromatography separation. Elemental analysis was carried out at the Micro-Analytical Unit (India).

#### 2.1.1. Synthesis of 3-(4-Acetylphenylcarbamoyl) Acrylic Acid (APA)

An equimolar amount of maleic anhydride (0.98 g) and 4-aminoacetophenone (1.37 g) was dissolved in 20 mL of acetone and stirred at room temperature. The crude APA began to precipitate as a light-yellow solid after stirring for 1 h. Stirring was continued for another 1 h and then 50 mL of ice-cold water was added. The crude precipitate was filtered, washed with cold acetone (20 mL × 3), then crystalized with hot ethanol, and dried under vacuum. The crystalized APA appeared as a light-yellow solid, which was then dried overnight at 60 °C. Yield: 94%; melting point: 220–222 °C. The ^1^H-NMR (400 MHz, DMSO-*d_6_*): 12.9 (s, *J* = 9.2 Hz, COOH), 10.6 (s, 1H, NH), 7.8 (dd, 2H, *J* = 8.8 Hz, ArH), 7.9 (dd, 2H, *J* = 8.4 Hz, ArH), 6.5 (d, 1H, *J* = 12 Hz), and 6.3 (d, 1H, *J* = 12 Hz); FTIR (KBr, cm^−1^): 3500, 3200, 2821,1666, 1626, 1545, 1415, 1300, and 850.

#### 2.1.2. Poly(3-((4-Acetylphenyl)carbamoyl)-2-ethyl-4-methylpentanedioic Acid) (PMA)

To a 100-mL, two-necked, round-bottom flask equipped with a spiral condenser and a magnetic stirrer, 2.33 g of APA was added. The flask was degassed by passing dry nitrogen for 30 min. AIBN (0.86 g) was dissolved in 10 mL of THF and injected into the reaction mixture. The whole reaction mixture was degassed again for another 30 min and constantly stirred in 65 °C. After 4 h, 50 mL of ethanol was added to the obtained colloidal product, and the resulting solid was filtered and washed with ethanol. The resulting polymer (PMA) was thoroughly washed with hot ethanol and dried under vacuum. 

Yield: 3.4 g (86%); ^1^H-NMR (400 MHz, DMSO-*d_6_*): 12.5 (s, 1H, COOH), 9.0 (s, H, NH), 7.67 (dd, *J* = 6.8 Hz, ArH), 6.56 (dd, 2H, *J* = 10 Hz, ArH), 3.7 (m, 2H, -CH-CH), and 2.6 (s, 3H, -COCH_3_-); FTIR (KBr, cm^−1^): 3395, 3222, 2821, 2354, 1643, 1591, 1396, and 1278.

#### 2.1.3. Poly 2-(1-((4-Acetylphenyl)amino)-1-oxobutan-2-yl)-4-phenylpentanoic Acid (PSA) 

Synthesis of PSA was preceded by the reaction of APA (2.33 g), styrene (1.04 g), and AIBN (0.25 mg). In a 100-mL, two-necked, round-bottom flask equipped with a condenser and a magnetic stirrer, which was maintained in a dry nitrogen atmosphere throughout the reaction, at room temperature, AIBN (0.25 mg) was dissolved in THF (10 mL), transferred to the flask containing APA (2.33 g), and degassed with dry nitrogen for 30 min. Then, styrene (1.04 g) in 10 mL of THF was injected into the reaction mixture with constant stirring. The reaction temperature was increased to 50 °C and stirred constantly. After 3 h, the reaction mixture with the crude PSA started to precipitate, where 50 mL of ethanol was added and allowed to reach room temperature. The crude product was filtered, washed with ethanol (20 mL × 3), and dried overnight in a vacuum oven. The molecular weight of the copolymer was determined by gel permeation chromatography (GPC) using THF. The synthesized PSA was characterized by spectral analysis (IR, ^1^HNMR, GPS, and UV-Vis). Yield: 3.75 g; ^1^H-NMR (400 MHz, DMSO-*d_6_*): 12.2 (s, 1H, COOH), 8.0 (s, 1H, NH), 7.67 (dd, 2H, *J* = 8.8 Hz), 6.0 (m, 5H, ArH), 6.57 (dd, 2H, *J* = 8.8 Hz), 3.5 (m, 2H, -CH-CH), and 2.7 (s, 3H, -COCH_3_-); FTIR (KBr, cm^−1^): 3182, 2353, 1652, 1593, 1526, 1401, 1273, 1668, and 920.

#### 2.1.4. Grafted Polymer 3-((4-(5-Amino-6-cyano-4-(2,4,6-trimethoxyphenyl)pyridin-2-yl)phenyl)carbamoyl)-2-ethyl-4-methylpentanedioic Acid (PMAP)

The synthesized PMA (5.0 g) was dissolved in benzene (50 mL) and then transferred to a 250-mL, round-bottom flask, equipped with a magnetic stirrer, and stirred at room temperature. To the above solution, 1,2,3-trimethoxy benzaldehyde (2.28 g), malononitrile (0.98 g), and ammonium acetate (6.3 g) were added and heated at 85 °C for 8 h. After completion of the reaction, the mixture was allowed to attain room temperature and the precipitated ammonium chloride was removed by filtration. A large volume of hot ethanol was added to the crude PMAP and the solution phase was filtered off to obtain purified PMAP. The purified PMAP was subjected to spectroscopic analysis. The ^1^H-NMR (400 MHz, DMSO-*d_6_*): 12.0 (s, 1H, COOH), 8.9 (s, H, NH), 7.72 (dd, 2H, *J* = 6.68 Hz), 6.66 (dd, 2H, *J* = 10Hz), 7.0 (s, 1H, ArH), 3.9 (s, 2H, OCH_3_), and 3.3 (m, 2H, -CH-CH); FTIR (KBr, cm^−1^): 3359, 2206, 1661, 1608, 1438, 1153, 1122, and 871.

#### 2.1.5. Grafted Polymer 3-((4-(5-Amino-4-(2-chloro-4,6-dimethoxyphenyl)-6-cyanopyridin-2-yl)phenyl)carbamoyl)-2-ethyl-4-methylpentanedioic Acid (PMAP1)

The synthesized PMA (5.0 g) was dissolved in benzene (50 mL) and then transferred to a 250-mL, round-bottom flask, equipped with a magnetic stirrer, where dissolution was achieved at 25 °C. To the above solution, 4-chloro benzaldehyde (2.1 g), malononitrile (0.98 g), and ammonium acetate (6.3 g) mixture was added and heated at 85 °C for 8 h. After completion of the reaction, the mixture was filtered off and the solid ammonium chloride was removed. A large volume of hot ethanol was added in the crude PMAP1 and the solution phase was filtered off to obtain pure PMAP. The purified PMAP1 was subjected to spectroscopic analysis. Yield: 2.7 g (43%); ^1^H-NMR (400 MHz, DMSO-*d_6_*): 12.0 (s, 1H, COOH), 8.9 (s, H, NH), 7.72 (dd, 2H, *J* = 6.68 Hz), 6.66 (dd, 2H, *J* = 10Hz), and 3.8 (m, 2H, -CH-CH); FTIR (KBr, cm^−1^): 3348, 2211, 1664, 1598, 1574, 1350, 1177, 860, and 474.

#### 2.1.6. Grafted Polymer 2-(1-((4-(5-Amino-6-cyano-4-(2,4,6-trimethoxyphenyl)pyridin-2-yl)phenyl)amino)-1-oxobutan-2-yl)-4-phenylpentanoic Acid (PSMP)

PSA (5.0 g) in benzene (50 mL) was placed in a 250-mL, round-bottom flask, and stirred at 25 °C. To the above mixture, 1,2,3-trimethoxybenzaldehyde (2.28 g), malononitrile (0.98 g), and ammonium acetate (6.3 g) were added, and then the reaction mixture was heated at 85 °C for 8 h. After completion of the reaction, the reaction mixture was filtered off and ammonium chloride was removed by filtration. Further, the crude PSMP was dissolved in hot ethanol and the solid product was collected by filtration. The purified PSMP was subjected to spectroscopic analysis. Yield: 2.1 g (31%). 

The ^1^H-NMR (400 MHz, DMSO-d_6_): 12 (s, 1H, COOH), 8.1 (s, 1H, NH), 7.9 (s, 9H, OCH_3_), 7.77 (dd, 2H, *J* = 8 Hz), 7.67 (dd, 2H, *J* = 8 Hz), 6.6 (m, 3H, ArH), 6.3 (m, 5H, ArH), 6.0 (s, 2H, NH_2_), 3.5 (m, 2H, -CH-CH-), and 2.9 (m, -CH_2_-CH); FTIR (KBr, cm^−1^): 3342, 2212, 1629, 1574, 1177, 1173, and 821.

#### 2.1.7. Grafted Polymer 2-(1-((4-(5-Amino-4-(2-chloro-4,6-dimethoxyphenyl)-6-cyanopyridin-2-yl)phenyl)amino)-1-oxobutan-2-yl)-4-phenylpentanoic Acid (PSMP1)

PSA (5.0 g) in benzene (50 mL) was placed in a 250-mL, round-bottom flask, and stirred at 25 °C. To the above mixture, 4-chloro benzaldehyde (2.1 g), malononitrile (0.98 g), and ammonium acetate (6.3 g) were added and then the mixture was heated at 85 °C for 8 h. After completion of the reaction, the reaction mixture was filtered off and ammonium chloride was removed by filtration. Further, the crude PSMP was dissolved in hot ethanol and the solid product was collected by filtration. The purified PSMP was subjected to spectroscopic analysis. Yield: 3.2 g (33%); ^1^H-NMR (400 MHz, DMSO-*d_6_*): *δ* 12 (s,1H, COOH), 8.1 (s,1H, NH), 7.88 (dd, 2H, *J* = 8.4 Hz), 7.68 (dd, 2H, *J* = 8 Hz), 7.62 (dd, 2H, *J* = 8 Hz), 7.0 (m, 2H, ArH), 6.8 (m, 5H, ArH), 6.1 (s, 1H, NH_2_), and 3.4 (m, 2H, -CH-CH-); FTIR (KBr, cm^−1^):3359, 2206, 1603, 1604, 1598, 1438, 1122, and 871.

## 3. Results and Discussion

### 3.1. Characterization of Grafted-Polymer Derivatives by FT-IR, NMR, and GPC Analysis

A series of 2-amino, 3-cyanopyridine polymers were synthesized in the presence of benzene by a one-pot synthesis via the Hantzsch reaction [21]. Copolymerization of MMA (maleamide-maleanilic acids) and ST (styrene) was carried out using ABIN as a catalyst. The PSA with molecular weights of 40,000 and 46,341 were for the polymer-grafting process. The composition of the grafted copolymers was analyzed using FT-IR/NMR spectroscopy and GPC analysis. FT-IR spectral results were used to detect the copolymerization of styrene-aryl amide. By comparing the FTIR spectra of copolymer [23] and PSA-(MA-*g*-ST-grafted polymer) PSMP (cf. Figure 1 and Figure 2), the shift of the IR band at 3344.6 cm^−1^ was due to the OH stretching vibration of the copolymer. The shift to 3338.8 cm^−1^ for MA-*g*-ST with lowering of the intensity is shown, which indicates the participation of hydroxyl groups during the chemical reaction. The FT-IR analysis, shown in Figure 2, further confirmed the grafting of pyridine on styrene-maleic acid (PSMP) with characteristic absorption bands of a strong band at 1666 cm^−1^ for CO (amide-I) and 1610 cm^−1^ for stretching of the NH_2_ group (amide- II). The band at 2208 cm^−1^was attributed to CN bond stretching and a broad and medium intensity peak at 734–640 cm^−1^ related to out-of-plane NH wagging vibrations (Figure 2).

The aromatic protons from grafted-PMAP and -PSMP fluorescent polymer compounds were identified by ^1^H-NMR spectra, as shown in Figure 3, Figure 4, Figure 5 and Figure 6. Figure 3 and Figure 4 show the ^1^H-NMR spectra of PMA and PSA, where both polymers had similar structural units and the hydrogen in each structural unit had comparable relaxation times. The ^1^H-NMR spectra of grafted-polymer PSMP (*δ* in ppm from in DMSO-*d*_6_) with the amide (NH) protons from PSMP showed a broad signature at 8.0 ppm. The peaks appeared at 7.76–7.75, 7.67–7.65 to 6.8–6.3 (doublets, aromatic protons) for the spectra of PSMP (Figure 5). The characteristic peaks of the 3-methoxyl group (3.9 ppm) are shown in Figure 5. The grafted polymers were assigned to the absorption of the -CH-CH- group within the styrene-maleic acid ring and the signatures at 3.8–2.0 ppm related to the maleic acid groups of styrene. As depicted in Figure 5 and Figure 6, the ^1^H group was identified as an NH attached to the phenyl ring that appeared as a singlet at 8.0 ppm and the (Ar-H) aromatic proton in the range of 6.5–6.3 with (Ar-Cl) aromatic protons appeared as doublets at 7.76–7.67 ppm and 7.68–7.66 ppm. 

### 3.2. Thermal Properties of Grafted-Polymer Derivatives

Thermal stability of synthesized polymer samples was characterized by using thermogravimetric analysis (TGA). Dehydration, decarboxylation, and depolymerization are the three stages of degradation that occur on grafted-copolymer pyridine derivatives (PSA). Figure 7 shows the TGA and DTG (derivative thermogram) of grafted polymers with two definite zones of weight loss. The weight loss relates to the liberation of water in the temperature range of 113–200 °C. The respective mass change was obtained around 18.1%. In the next stage, the weight loss in the temperature range of 311–575 °C was observed around 44.8%, due to further liberation of CO_2_ and other carbon derivatives of the framework [26,27]. On further heating, the complete decomposition of polymer occurred at 954 °C at the third stage, with a final weight change of 26.3%. The presence of appreciable events near 161 °C, 451 °C, and 954 °C in the DTA profile clearly showed the thermal stability pattern for synthesized poly (styrene-maleic acid-*g*-maleamide) PSA polymer (cf. Figure 7).

The thermal behavior and the weight loss profile of the grafted copolymers (PSMP1) are shown in Figure 8. The as-synthesized samples (10 mg) were placed in a silica crucible and heated up to 800 °C at a rate of 10 °C/min in an inert nitrogen atmosphere. The TG and DSC profiles of the graft polymers (PSMP) showed typical behavior of the decomposition process for such types of grafted copolymers (Figure 8). The grafted polymers showed very good thermal stability up to 300 °C and the respective DSC profile showed an endothermic event at 200 °C. 

The as-shown grafted polymer (PSMP) is promising for its more stabilized structure, and the further increased heating process resulted in the gradual decomposition of the polymer up to 900 °C. Compared to PSA, the grafted PSMP did not show any loss of grafted organic groups on the framework up to 300 °C, where resistance toward decomposition was evidenced by the grafted groups due to the stable bonding. The presence of the pyridine group anchored in PSMP and the number of methoxy groups anchored in PSMP may account for the enhanced thermal stability of the prepared, grafted-polymer derivatives during the thermal treatments [25,26,27,28]. 

### 3.3. Determination of Molecular Weight by GPC Analysis

In polymer characterization, molecular weight determination is a basic parameter that can be obtained reliably by gel permeation chromatography (GPC). The number average molecular weight (M_n_) and the weight average molecular weight (M_w_) of the polymer samples were determined using a Waters 510 HPLC pump equipped with a Waters 410 differential refractometer instrument at an operating wavelength (λ = 930 nm). The GPC instrument has three Waters Styragel columns, *viz.*, HR1, HR2, and HR3, with a size of 7.8 × 300 mm. Initially, the calibration techniques were done according to the standard method with poly(styrene) standards to avoid potential interference of the freely existing acidic compounds in the synthesized pyridine compounds. The GPC experiment was carried out at 40 ± 1 °C using HPLC grade acetonitrile (CH_3_CN) as the mobile phase at a flow rate of 1.0 mL/min. 

The average molecular weights (M_n_ & M_w_) and polydispersity index (PDI = M_w_/M_n_) of the synthesized polymers were calculated and the results are presented in Table 1. The PDI of compounds PSMP (M_w_ = 5.15 × 10^4^, PD = 3.4), PSMP1 (Mw = 5.22 × 10^4^, PD = 3.3), and PMAP1 (Mw = 5.23 × 10^4^, PD = 3.0) were found to be high. The obtained, large PDI values (PDI > 3.00) of pyridine-acrylic acid–styrene polymers were due to fragmentation and recombination of the monomers. The branched structures may have been the reason for the large PDI values of the polymers. Furthermore, the other polymers had PDI values > 2 (PDI = 2.6 to 2.8), which indicated that the synthesized polymers had a cross-linked network and hyper-branched chain lengths. It was possible to calculate the degree of branching in the polymer backbone using NMR analysis of the terminal groups present in the polymers. Similarly, the FTIR spectra of the monomers in the reaction could be used to study the presence of functional groups in the polymers, accordingly.

### 3.4. Absorption and Fluorescence Spectroscopy Characterization

#### Absorbance Spectra

The UV-vis absorption spectra of fluorescent pyridine analogues of PSA, PSMP, and PSMP1 samples were recorded using dimethyl formamide (DMF) as solvent due to the higher solubility of the polymers in DMF. The maleamide copolymer exhibited fluorescent properties that caused the florescent emission of the poly (styrene-4-acetyl-maleic acid-arylamide) (PSA) polymer (cf. Figure 9). The maleic acid-acrylamide units of PSA were a type of electron-deficient monomer that usually co-polymerizes with styrene monomers. Herein, the 3-(4-acetylphenylcarbamoyl) acrylic acid was co-polymerized with styrene monomers to prepare polymers that contain maleic acid moieties, and the reactions occurred smoothly using AIBN as a radical initiator. The graft polymer PSMP, graft PSMP, was synthesized by employing poly(styrene-4-acetyl-maleic acid-arylamide) (PSA). The copolymer PSA, maleimide copolymer, had some fluorescent properties, where the poly(styrene-maleimide) (PSA) arylamide units were a kind of electron-deficient polymer system. In the UV absorbance spectra herein, a maxima for the co-polymer was observed at 330 nm (PSA, λ_max_ 330 nm), as shown in Figure 9. The emission spectra showed a λ_em_ near 240 nm and the emission spectra of the PSA polymer revealed a blue emission in other solvents such as CHCl_3_, DMF, and DMSO. The PSA polymer exhibited a broad absorption and emission band at 313–500 nm (λ_max_ (absorbance) = 375 nm). 

For the spectral measurements, PSA and PSMP were prepared at 1 × 10^−6^ M in DMF (HPLC grade), and fluorescence spectra were recorded at room temperature. It was evident that the photochemical properties of the fluorescent pyridine-based copolymers were influenced by the nature of the substitutions present in the aromatic ring [20]. All the prepared copolymers contained pyridine polymer derivatives, which were UV active, and the absorption wavelength λ_abs,max_ was observed between 335 and 340 nm, whereas the DMF solvent was colorless to the naked eye. The blue shift of the absorbance may have been due to the pyridine substitution with the core dye subunit, styrene-maleimide. In general, a pyridine compound shows stronger n-π* transition than a π-π* transition due to the presence of extended π conjugation system in the structure. The excited-state behavior of the synthesized, grafted pyridine containing copolymer analogs (PSMP derivatives) was obtained. Figure 10 shows the fluorescent emission spectra of PSMP and PSMP1 in DMF solution (λ_em,max_ = 300–600 nm). 

Zhao and his co-workers reported the preparation of an emissive polymer from a non-fluorescent monomer [22]. In their study, polymers such as styrene and vinyl amide were utilized as monomers for the interaction between the carbonyl group and the phenyl ring in the polymer, which are the key ingredients for the observed light emission with a maximum intensity at 330 nm in DMF solution. The absorption spectra of the maleimide-styrene copolymer were checked in various solvents, where they were freely soluble in DMF and DMSO at a polymer concentration of 10^–6^ M. In Figure 10a,b, it is clear that the small shoulder for the absorbance band obtained for PSMP1 near 550 nm compared to PMAP and PMAP1, where additional absorption occurred due to the functional group present in the PSMP1. In the case of the fluorescence spectra of PSMP and PSMP1, the grafted PSMP1 showed the reduced peak intensity, with emission peaks at 300 nm and 365 nm, compared to PSMP (cf. Figure 10a). Hence, it was confirmed from the fluorescence results that the excitation and emission spectra of the PSMP and PSMP1 polymer indicated a blue emission band [23]. 

The grafted copolymer had a structure similar to PSA, as per the photoluminescence studies of the two polymers (λ_max_ 330–316 nm for PSMP1, and PMAP1). The PSMP1-grafted polymer displayed a broad emission band at 400–500 nm (maximum wavelength 460 nm) [24,25,26,27,28]. The highest emission intensity was observed for PSMP and PSMP1 upon excitation by irradiation of the blue region. Figure 11 shows the absorption and fluorescence spectra of the homopolymers (PMAP, PMAP1) that showed similar absorption maxima (λ_max_ = 450 nm). There were no other shoulder bands observed from PMAP and PMAP1, as compared to the PSMP1-grafted polymer. The fluorescence emission peak for both PMAP and PMAP1 revealed that the λ_em_ values were 360 nm and 335 nm, respectively. The additional functional groups in PMAP caused a reduction in the fluorescence peak intensity. The role of the solvent played an important role in color formation of the polymer and the grafted forms. According to the results summarized in Figure 9, Figure 10 and Figure 11, similar solvent effects on the emission spectra of PMAP and PMPA1 were observed. Compared with PSA, the solutions of PSMP and PSMP1 were more intensely colored in the same solvent. The smaller number of pyridine groups in PSMP and large steric hindrance imposed by the methoxy group in PSMP may have affected the extent of their interaction with the solvent, thus enhancing the changes in absorption and emission. The fluorescence lifetime spectra of all synthesized polymers are shown in Figure 12. The fluorescence lifetimes (2.57–6.5 ns) indicated that the grafted polymer had excitation of a proton to a higher energy state that was favored by the presence of the rigid molecular structure in the prepared polymers [29,30,31,32]. Figure 12a,b shows differences in the lifetime of emission and excitation of PSA and PSMP. The other prepared polymers, PSMP1 and PMAP (cf. Figure 12c,d), also showed a higher lifetime for excitation of protons to higher energy compared to PSA and PSMP [28].

### 3.5. Antimicrobial Activity of Cyanopyridine-Based Polymer Derivatives 

#### In Vitro Antibacterial Activities

According to previous reports, there are examples of pyridine analogues with variable antimicrobial activities. By contrast, the newly synthesized fluorescent cyanopyridine-modified styrene copolymer analogues and grafted polymers have not been reported to date for their structure–property relationships related to antibacterial and antifungal activity. The present study focused on evaluating the potential biological activity of such systems for the health sector and infection-control applications. The greater microbiocidal activity relates to high cationic charge of the amide groups in the polymer derivatives, along with their hydrophobicity due to alkyl chain-length effects. The as-prepared modified cyanopyridine polymers possessed variable hydrophile-lipophile balance (HLB) that contributed to their high activity and relative water insolubility. In turn, the structure−property–activity relationships of the PSA, PSMP, PSMP1, and PMAP were evaluated since variable antimicrobial activity was anticipated for systems that had variable molecular weights, molecular architecture, and HLB. Further, the antimicrobial activity of modified polymers has been reported mainly for bacteria, despite the prominence of fungal infections as one of the key hazards to human health. Moreover, fungal biofilms are known to allow the formation of bacterial colonies on their surface and serve to enhance the bacterial resistance to antibiotics. Hence, it is meaningful to develop polymers with both antibacterial and antifungal activity, along with studies of their structure−activity relationships, for applications as thin film coatings for the textile industry and health care sectors. In the section below, the propensity of bacteria to develop resistance against the as-prepared grafted polymers was studied against both Gram-positive and Gram-negative bacteria. 

Herein, the prepared fluorescent polymers were subjected to antimicrobial activity against *S. aureus* MTCC 25923 and *E. coli* MTCC 25922 pathogens, where tetracycline was used as a positive control. PSMP was effective in controlling both the pathogens with a high zone of inhibition at a concentration of 50, 75, and 100 μg/mL. However, it is interesting to note, at higher concentrations (100 μg/mL), the zone of inhibition was similar to the positive control. PSA (11–21 mm), PSMP1 (7–9 mm), and PMAP (10–26 mm) showed affected zones of inhibition for both bacterial strains (cf. Table 2). The antifungal activities of the as-prepared pyridine polymers were evaluated using *C. albicans* and *A. niger* fungal pathogens. The prepared PSA, PMAP, and PSMP showed the effective zone of inhibition for *C. albican* pathogens at higher and lower concentrations. In the case of fungal pathogen *A. niger*, the pyridine-based polymers (except PSMP and PSMP1) showed an effective zone of inhibition related to control of Carbendazim. The higher anti-fungal activities observed for PSA, PSMP, and PMAP pyridine polymers for pre- and post-grafted bulky polymer structures hindered the active sites present in the pyridine derivative, which may account for the lowered activity of grafted polymers such as PSMP1 and PMAP-1. Figure 13 and Figure 14 show an illustrative bar graph of the activity ratio for as-prepared pyridine polymers for antibacterial and anti-fungal activities [26,32,33]. The schematic expression of *E. coli* bacteria in the presence of the control (tetracycline) and the activity of cell expression of the synthesized pyridine polymers for bacterial activity is shown in Figure 14.

## 4. Conclusions

The copolymerization of PMA and PSA was carried out using AIBN as the radical initiator. The acetyl group of the copolymers (PMA/PSA) was used for grafting of poly(styrene-maleic acid-*g*-methoxy pyridine) by the Hantzsch reaction using substituted benzaldehyde, manonitrile, and ammonium acetate. All synthesized polymers were characterized unambiguously by FT-IR/^1^H-NMR spectroscopy and GPC. The thermal analysis results confirmed that the polymers were thermally stable up to 900 °C. Compared with PSA, the thermal stability of PSMP improved substantially due to the grafting process. The UV absorbance of the PSA co-polymer (λ_max_ 330 nm) in the excitation and emission spectra showed a blue emission in DMF. The fluorescence-emission spectra showed that the copolymer (PSA) absorbed light at a specific frequency and it emitted at the same frequency. The pyridine and styrene moieties showed strong aggregation-induced emission in the poly(styrene-4-acetyl maleic acid-arylamide) copolymer. The emission of PSA was associated with the molecular interactions between the carbonyl group, and its pyridine backbone was determined via fluorescence spectroscopy results. The synthetic polymers (PSA and PSMP1) displayed very strong, blue emission in DMF, which was observed under UV light with fluorescence emission of polymers (460 nm) with a maximum emission at 375 and 425 nm using fluorimetry. Further, the synthesized polymers were subjected to antimicrobial studies that showed a remarkable zone of inhibition toward all types of pathogens. In particular, PMAP and PSA showed maximum inhibitory activities against *C. albicans* and *A. niger* at both lower and higher concentrations. 

## Data Availability

The data presented in this study are available on request from the co-corresponding author (J.R.). The data are not publicly available because the raw/processed data required to reproduce these findings cannot be shared at this time as the data also form part of an ongoing study.

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
