# Peer review of "Synthesis and Characterization of Pyridine-Grafted Copolymers of Acrylic Acid–Styrene Derivatives for Antimicrobial and Fluorescence Applications"

_micromachines, 2021, doi:10.3390/mi12060672_

Round 1

Reviewer 1 Report

  The only suggestions I have are as follows:

  1. The numbers of significant figures in Table 1 are excessive
  2. It is stated that C-13 NMR was used to characterize the copolymers  but no results were presented. Too bad, because they may have allowed the determination of co-monomer and stereo-sequences.  

Author Response

Authors’ Response to Reviewer Queries on MS ID: micromachines-1228336

Reviewer #1

The only suggestions I have are as follows:

  1. The numbers of significant figures in Table 1 are excessive

Authors’ Response:

In Table 1, the observed numerical data has now been rounded to the nearest significant number in the revised manuscript and the corresponding discussion has been modified based on the significant values.

Corrected Table:

Table 1. Molecular weight values (Mn, Mw) and PDI of the prepared terpolymers PSA, PSMP and PSMP1.

Polymer

Material

Weighted Average Values

Mw (g/mol)

Mn (g/mol)

PDI

PSA

4.63´104

1.63´104

2.83

PSMP

5.15´104

1.52´104

3.37

PSMP1

5.22´104

1.57´104

3.31

PMA

3.73´104

1.44´104

2.58

PMAP

4.47´104

1.61´104

2.77

PMAP1

5.23´104

1.72´104

3.04

The average molecular weights (Mn & Mw) and polydispersity index (PDI = Mw / Mn) of the synthesized polymers have been calculated and the results are presented in Table 1. The PDI of compounds PSMP (Mw = 5.14´104, PD = 3.4), PSMP1 (Mw = 5.22´104, PD = 3.3), and PMAP1 (Mw = 5.23´104, PD = 3.0) were found to be high. The obtained large PDI values (PDI > 3.00) of pyridine-acrylic acid–styrene polymers are due to fragmentation and recombination of the monomers. The branched structures may be the reason for the large PDI values of the polymers. Further, all of the other polymers have PDI values > 2 (PDI = 2.6 to 2.8), which indicate that the synthesized polymers have a cross-linked network and hyper branched chain lengths. It is possible to calculate the degree of branching in the polymer backbone using NMR analysis of the terminal groups present in the polymers. Similarly the FTIR spectroscopic analysis, completion of the monomers in the reaction can be used to study by analyzing the monomers functional groups via their IR spectral results.

The abstract was revised accordingly.

      So for, we did not discussed about carbon NMR in the discussion.  The statement presented in the materials section about the carbon NMR has now been modified in the revised manuscript. Ofcourse the NMR studies used for determine monomer reactivity, we used TLC because of organic compounds. Also, the FTIR did not show the presence of monomer in the polymers. 

Authors’ Response: Page 3, section 2.1 Materials: Remove “and 13C-NMR”

As well, the authors wish to acknowledge Reviewer #1 for the constructive and critical comments on this manuscript, along with the opportunity to make improvements.  We have further edited the entire manuscript for syntax, language, and clarity throughout to meet the high standards of this journal.

Reviewer 2 Report

Introduction

Introduction was mostly well written, covered the main background points and led up to the aim of the study. 

Methods

Authors presented the key element of the study design and adequately described the used methods. 

Results and discussion

There is an issue with the figures 1, 7 and 8. It is difficult to read results on these graphs because the images are uploaded in very low resolution. I suppose that these images are exported directly from instruments and that authors could not export them in higer resoulution. Therefore, I suggest authors to insert the most important values on these figures in other program like the most simplest paint like they have already done in Figure 7.

Equation (1) and (2) are just inserted into the text without any explanation of what they represent. It is more usual if authos insert something in the text that are not words that they name it like figure xy or picture xy. Therefore, I  I suggest you write an explanation below the picture/figure, because it is acutally a picture.

Overall, this is an interesting study in which a large number of experiments was made and authors have used a lot of sophisticated analytical instruments. Also, authors have tested in vitro antimicrobial activity of these copolymers which increases the interest and significance of this study.

Author Response

Authors’ Response to Reviewer Queries on MS ID: micromachines-1228336

Reviewer #2

Introduction

Introduction was mostly well written, covered the main background points and led up to the aim of the study. 

Methods

Authors presented the key element of the study design and adequately described the used methods. 

Results and discussion

There is an issue with the figures 1, 7 and 8. It is difficult to read results on these graphs because the images are uploaded in very low resolution. I suppose that these images are exported directly from instruments and that authors could not export them in higer resoulution. Therefore, I suggest authors to insert the most important values on these figures in other program like the most simplest paint like they have already done in Figure 7.

Authors’ Response:

Thanks for the suggestion, we modified Fig. 1, 7 and 8 for more clarity and the values are clearly visible to the readers.

Equation (1) and (2) are just inserted into the text without any explanation of what they represent. It is more usual if authos insert something in the text that are not words that they name it like figure xy or picture xy. Therefore, I  I suggest you write an explanation below the picture/figure, because it is acutally a picture.

Authors’ Response: Equation 1 and 2 is general representation of fluorescence lifetime spectra and we removed from the revised manuscript due to commonly known equation.

Overall, this is an interesting study in which a large number of experiments was made and authors have used a lot of sophisticated analytical instruments. Also, authors have tested in vitro antimicrobial activity of these copolymers which increases the interest and significance of this study.

Authors’ Response:

Authors express their thanks to the reviewer for this comment about our research work. We are interest to further explore biomedical applications of high efficacy compounds in the near future.

As well, the authors wish to acknowledge Reviewer #2 for the constructive and critical comments on this manuscript, along with the opportunity to make improvements.  We have further edited the entire manuscript for syntax, language, and clarity throughout to meet the high standards of this journal.
